# Remaining-Useful-Life Prediction and Uncertainty Quantification using LSTM Ensembles for Aircraft Engines

**Oishi Deb**
Department of Engineering Science
University of Oxford, UK
oishideb@robots.ox.ac.uk

**Emmanouil Benetos**
School of Electronic Engineering
and Computer Science
Queen Mary University of London, UK
emmanouil.benetos@qmul.ac.uk

**Philip Torr**
Department of Engineering Science
University of Oxford, UK
philip.torr@eng.ox.ac.uk

## Abstract

This paper proposes "LSTM (Long Short Term Memory) Ensemble" technique in building a regression model to predict the Remaining-Useful-Life (RUL) of aircraft engines along with uncertainty quantification, utilising the well-known run-to-failure turbo engine degradation dataset. This paper addressed the overlooked yet crucial aspect of uncertainty estimation in previous research, by revamping the LSTM architecture to facilitate uncertainty estimates, employing Negative Log Likelihood (NLL) as the training criterion. Through a series of experiments, the model demonstrated self-awareness of its uncertainty levels, correlating high confidence with low prediction errors and vice versa. This initiative not only enhances predictive maintenance strategies but also significantly improves the safety and reliability of aviation assets by offering a more nuanced understanding of predictive uncertainties. To the best of our knowledge, this is a pioneering work in this application domain from a non-Bayesian approach.

## 1 Introduction

Predictive maintenance, harnessing machine learning for timely upkeep, has become crucial in engineering and manufacturing, notably reducing costs and enhancing revenue through early aircraft engine degradation detection and accurate Remaining-Useful-Life (RUL) predictions [1].

While past RUL prediction research for turbo engines, such as [2] and [3], has primarily utilized logistic regression and standard Artificial Neural Networks (ANN), a significant gap remains in exploring predictive model uncertainty, especially in safety-critical systems [4, 5, 6, 7]. Deep learning models, despite their varied successes, often display overconfidence in predictions, emphasizing the need for accurate uncertainty estimation [8, 9].

This paper addresses this gap, developing a regression model using "LSTM Ensemble" to predict turbo engine RUL, leveraging the widely recognized turbo engine degradation dataset [10]. Metrics like RMSE, $R^2$, and Negative Log Likelihood are employed, with the paper aiming to enhance the Deep Ensemble technique by estimating Overall-Predictive and Epistemic uncertainties, offering

37th Conference on Neural Information Processing Systems (NeurIPS) Workshop on Advancing Neural Network Training (WANT): Computational Efficiency, Scalability, and Resource Optimization 2023.

a comprehensive understanding of predictive uncertainty. Further explanation on Epistemic and Overall-Predictive Uncertainty computation is given in Appendix.

## 2 Literature Review

### 2.1 Uncertainty Quantification methods

Uncertainty pertains to a state of ambiguity, and this is a phenomenon that machine learning models occasionally grapple with in relation to their predictions. Statistical methodologies, such as Confidence Intervals and Monte Carlo Simulations can be used for gauging this uncertainty.

A Confidence Interval provides a specified range within which the predicted value is likely to fall, as elucidated by Cosma Shalizi [11]. In practice, if a Neural Network model undergoes multiple iterations, and the standard deviation of the output values is calculated, it becomes feasible to deduce the confidence interval in an offline manner. Nonetheless, given that Neural Network models may comprise millions of parameters, executing them repeatedly incurs substantial computational costs. To mitigate this, the strategy of ensembling multiple Neural Networks, also known as Deep Ensemble, was introduced during the training phase. A comprehensive discussion on Deep Ensemble is provided in a subsequent section.

It is crucial to highlight that calculating the confidence interval offline to ascertain uncertainty is fundamentally different from deriving uncertainty directly from the model. When a deep learning model proactively provides an estimate of uncertainty alongside its prediction, it demonstrates an intrinsic awareness of its confidence level. This self-awareness is of paramount importance, especially in safety-critical applications such as autonomous systems, where understanding and acknowledging the model's limitations and uncertainties is essential.

As previously highlighted, conventional Neural Network architectures do not inherently provide an estimate of uncertainty in their predictions. This necessitates a modification and enhancement of existing Neural Network structures to integrate statistical methodologies such as Confidence Intervals and Monte Carlo Simulations, facilitating the extraction of uncertainty from the model.

In the realm of deep learning, the most prominent methods for uncertainty estimation encompass Bayesian and non-Bayesian techniques. Monte Carlo Dropout, a Bayesian technique, stands out as a significant method and is thoroughly discussed in the work of Gal et al. [12]. On the other hand, Deep Ensembles, a non-Bayesian approach detailed by Balaji [8], is well-known for its computational efficiency. While there are various other methods available for uncertainty estimation mainly from Bayesian point of view, as outlined in Table 6, the paper at hand has opted to employ Deep Ensembles for being the most state-of-the-art model due to its computational efficiency.

**Monte Carlo Dropout:** Probabilistic machine learning encompasses Frequentist and Bayesian strands, with the Monte Carlo dropout technique aligning with the latter [13]. While dropout was initially explored by [14] and [15] from a Bayesian perspective, [12] enhanced it, demonstrating its use as a Bayesian Approximation to integrate over Neural Network model weights. Unlike traditional Neural Networks, Bayesian Neural Networks initialize weights with prior probability distributions, commonly Gaussian, and compute posterior distributions instead of point estimates.

Introduced by [16] to mitigate overfitting, dropout was traditionally used during training. However, [12] applied it during both training and testing, yielding non-identical outputs during test time with different dropout values, analogous to Monte Carlo (MC) sampling [12, 17]. This variance in output samples provides uncertainty estimates. Although MC Dropout is a recognized method for uncertainty quantification from a Bayesian standpoint, the subsequent introduction of a non-Bayesian method, 'Deep Ensemble,' by DeepMind [8] offered an alternative computationally efficient approach, detailed in the following section.

The limited industrial deployment of the MC dropout method is attributed to the computational expense and scalability challenges of Bayesian Inference [8]. In a more recent industrial research study conducted in 2021, MC Dropout was experimented with for Smart Grid Design application [18], however, it has not been deployed fully yet.

**Deep Ensembles:** Deep Ensembles, a probabilistic model, which is also known as ensembles of neural networks are not as computationally heavy as Bayesian models as they do not use Bayesian

Inference. The concept of the ensemble has been widely used in the regularisation of Neural Network [19], however, this method also proves useful in uncertainty estimates as mentioned in [8].

In probabilistic regression models, the output is considered to be a Gaussian or Normal distribution with parameter mean and variance [20], hence this Distribution is used in calculating the Negative Log Likelihood (NLL). This NLL is then used as a cost function for the Deep Ensembles model. The Deep Ensembles method implements the concept from [21] in which it modifies the neural network to output the predictive mean and the standard deviation in the output layers, as opposed to classical neural networks for regression tasks where it only outputs a point estimate of the prediction. Confidence intervals can be calculated from the standard deviation to interpret the uncertainty in the prediction model.

Below is the equation of the cost function - Negative Log Likelihood (NLL). The full derivation of the equation is shown in the Appendix.

$$NLL = 0.5 \log(2\pi\sigma^2) + 0.5\sigma^{-2} \sum_{n=1}^{N} (x_n - \mu)^2 \tag{1}$$

where - $\sigma^2$: is the variance of the Gaussian distribution; $\sigma$: is the standard deviation of the Gaussian distribution; $\mu$: is the mean of the Gaussian distribution; $x_n$: is the individual data-points from 1 to N. $N$: is the total number of data points.

The NLL equation in the Deep Ensemble paper [8] removes the $2\pi$ in the first term and adds an additional constant term. Since the aim is to minimise the cost function and as $2\pi$ is a constant hence removing this term will not make any difference in the minimisation. An additional constant term was added in the equation as mentioned in [8], which is usually done when working with logarithms to avoid underflow and overflow errors. Negative Log Likelihood is used because it is a proper scoring rule that is widely used for evaluating predictive uncertainty [8]. The proper scoring rule is a function of the probability prediction and the output variable, and this function is minimum when the prediction is well-calibrated [11].

In other words, NLL is a way of measuring the error of the predictions. This works for models that predict sufficient statistics over some distribution (for example mean and variance for a Gaussian in this case) and then NLL can be calculated for the ground truth under the predicted distribution. The reason this metric is used in models that use uncertainty is because the model predicts the variance/standard deviation and uses it for the NLL, so the prediction of the uncertainty will affect this metric. However, variance/standard deviation is merged with the prediction of the mean, so just from the NLL it is not possible to make any claims about the uncertainty itself.

## 3 Procedure

### 3.1 Dataset

An open-source dataset [10] has been used which is available from the NASA repository. The dataset is run-to-failure turbo engine data created using the Commercial Modular Aero-Propulsion System Simulation (C-MAPSS) [22]. C-MAPSS was developed by NASA which simulates the real-life controls system for 90,000 pound thrust dual spool and high-bypass ratio turbo engines. This simulation environment was developed to facilitate research in control systems, Engine Health Monitoring (EHM), predictive maintenance etc. The dataset contains data from 4 different fleets of engines and each of the fleet data and information on the number of engine units are on Table 1. Table 2 shows the list of sensor names whose values are recorded for each engine unit in the dataset for each cycle until the engine fails. An engine cycle includes three events: engine start, take-off and landing, and engine shutdown. Take-off and landing are classed as a single event because take-off is always followed by a landing.

Further information on the dataset, data pre-processing and data labelling is provided in the Appendix.

## 3.2 LSTM (Long Short-Term Memory) Ensemble Model Building

We proposed "LSTM Ensemble" model which is based on the initial work of the Deep Ensemble [8] technique. We implement LSTM network architecture model with an input size of sequence length and number of features. According to the dataset [10] there are 25 input features - which are cycle number, 3 operating settings, and 21 sensor values.

### 3.2.1 Justification for using LSTM Ensemble Model

LSTM Ensemble has been used over other methods such as ARIMA [23], GARCH [24], or even simpler RNNs [25], is because ARIMA and GARCH are primarily linear models and might not capture non-linear patterns effectively, given the fact that our dataset is non-linear. While simple RNNs can handle non-linearities, they are not as efficient as LSTMs in learning from complex, high-dimensional data. LSTMs, being a more advanced form of RNNs, can model complex non-linear relationships in data much better.

Moreover, LSTMs are more robust to noise and non-stationarity in data compared to ARIMA and GARCH. Although ARIMA can handle some level of non-stationarity, LSTMs, especially when used in an ensemble setting, can better adapt to changing patterns in the data without the need for stringent pre-processing like stationarity tests and transformations.

While GARCH models are specifically designed to model volatility clustering and heteroskedasticity in financial time series, LSTMs can also capture these features, especially when trained on large datasets. The ensemble approach further enhances this capability by aggregating insights from multiple models. The Deep Ensemble approach, when combined with LSTMs, can lead to improved prediction accuracy. Each model in the ensemble will capture different aspects of the data, and their aggregation leads to a more accurate and robust model. This is particularly useful in our scenarios because the data contains complex patterns that a single model will not be able to capture effectively.

By integrating LSTMs, which effectively capture temporal dynamics, with Deep Ensembles, the model not only predicts future values but also gives a more reliable estimate of the uncertainty associated with these predictions. This is crucial for decision-making processes in fields where understanding the confidence in predictions is as important as the predictions themselves.

Finally, LSTMs, especially when implemented with efficient computational frameworks, can be scaled for large datasets and real-time analysis, which is a challenge for traditional models like ARIMA and GARCH which require extensive computations for each new prediction.

### 3.2.2 Training and Inference

The dataset, divided into training and testing files, allocated ten percent of the training data for validation. Table 8 displays the finalised parameter and hyperparameter values, determined after numerous experiments detailed in Table 9. The hyperparameters are tuned/optimized using grid search. Dropout, a regularization technique that probabilistically omits inputs during network training to mitigate overfitting, is applied after each LSTM layer. We implemented a grid search where several models are trained with different dropout rates (such as 0.1, 0.2, 0.3, ..., 0.7) and compared their performance on a validation set. 0.2 is found to be the most optimal value for the dropout rate.

It is crucial to note that the dropout used for regularization here differs from Monte Carlo Dropout [9], utilised for uncertainty quantification. The model's architecture is given in Table 7. The model is developed using Python's Keras [26] (version 2.3.0) with TensorFlow [27] (version 1.14.0) as a backend. The entire code is run on CPUs so it is computationally affordable. The model employed "Keras Uncertainty" open-source resources [28] for the Deep Ensemble model creation.

The Ensemble model was implemented using a custom Class, which takes in two arguments: first a function that creates a Neural Network LSTM model and second the number of neural network models the user wants in the ensemble. The Neural Network model provides two outputs: prediction value and the standard deviation. Since the standard deviation output of each member in the ensemble does not have direct supervision, unlike the prediction value (i.e. the prediction value is supervised by the target $y$), the standard deviation is indirectly supervised by the loss function NLL. This has been implemented by passing the standard deviation output to the loss, in which case, the training model does not output the standard deviation directly, but it is included in the loss so it influences the loss correctly.

# 4    Evaluation and Discussion

## 4.1    Evaluating Model with Uncertainty Quantification

The LSTM Ensemble model, tested on four-engine fleets, exhibited the highest error rate with the FD004 dataset, prompting numerous experiments detailed in Table 9 to enhance performance. The row highlighted in blue indicates optimal performance in both prediction and uncertainty estimates. While augmenting the number of Neural Networks (NN) in the ensemble improves performance, an ensemble exceeding three NN elevates the error rate. Upon identifying the top-performing model using the FD004 dataset, it was applied to the other three fleet datasets. Table 10 reveals the best RMSE value as 30.65 for the "FD001" dataset, with a Overall-Predictive standard deviation (uncertainty) of 42.46, placing the prediction value within a $\pm 42.46$ range of the mean prediction for this dataset. An $R^2$ of 0.53 for the FD001 indicates a commendable fit.

Figure 1 displays the prediction graph, with green denoting ground truth and blue representing prediction, figure 2 illustrates model prediction uncertainty with black and red trends. Ideally, the model should exhibit minimal uncertainty to maximize prediction certainty. However, establishing an uncertainty value threshold or conducting further comparisons necessitates the availability of additional baseline results for analysis.

**NLL for evaluating uncertainty**: The "FD001" dataset yielded a minimum NLL value of 3.95, which, while not a direct measure of uncertainty, facilitates the evaluation of the model's standard deviation during loss function minimization, thereby considering both prediction and its associated uncertainty. In the context of the four datasets, a lower NLL not only signifies more accurate predictions but also reliable uncertainty values. Overall-Predictive uncertainty suffices for assessing a predictive model's suitability, as demonstrated by [8]. However, incorporating Epistemic uncertainty provides nuanced insights from individual Neural Network (NN) models, enhancing decision-making.

For instance, the NN model reflects its confidence in its predictions. In the "FD002" fleet dataset, a high RMSE of 39.69 corresponds with a high Overall-Predictive standard deviation (uncertainty) of 63.62, indicating the model's awareness of its imperfect prediction. Low uncertainty could be perilous, signaling overconfidence in the model. In the "FD002" fleet dataset, it has the most units (260 engines) and the lowest Epistemic standard deviation (33.41) among the four fleet datasets, implying a desirable decrease in Epistemic standard deviation with increasing dataset size, the Overall-Predictive uncertainty remains elevated due to a high error rate. Regarding the "FD004" dataset, it presents an RMSE of 41.91 and a substantial Overall-Predictive standard deviation of 57.97, revealing the model's cognizance of its prediction's limited reliability. While inaccurate, the model's lack of confidence is non-fatal. The absence of an uncertainty value could be hazardous in safety-critical applications, as it leaves users blind to the model's confidence level.

Note that determining the threshold for uncertainty in maintenance decisions is context-dependent and involves risk assessment by the organisations. This paper does not detail specific thresholds but highlights that lower NLL values indicate more reliable uncertainty estimates. Maintenance teams might set thresholds based on acceptable risk levels and the criticality of engine components.

# 5    Conclusion and Further Work

Previous studies on predicting the Remaining Useful Life (RUL) of turbo engines did not address the uncertainty of the predictive model in safety-critical applications. Given the state-of-the-art deep ensemble model, the paper proposes "LSTM Ensemble" a novel approach to address the gap in the literature by implementing the Deep Ensemble method with LSTM to quantify model uncertainty. Significantly, it marks the first integration of LSTM architecture with Deep Ensemble for RUL predictions and uncertainty assessments in aircraft engines, utilizing NASA's engine degradation dataset.

While the LSTM Ensemble method showcased computational efficiency and delivered acceptable results on the FD001 and FD003 datasets, with Root Mean Square Error (RMSE) values of approximately 30 and 33, it encountered challenges with the FD002 and FD004 datasets. These datasets presented larger sizes and higher levels of data noise, resulting in elevated error rates.

This paper lays the groundwork for future research, underscoring the need to augment existing predictive models with methods for providing uncertainty estimates. Prospective directions include

efforts to reduce predictive uncertainty and leverage uncertainty estimates to mitigate generalization error. Additionally, uncertainty estimates could be instrumental for Out-Of-Distribution (OOD) detection [29, 30], signaling increased uncertainty for predictions related to values outside the training set's scope.

Investigations into advanced sequence models, such as Encoder-Decoder architectures [31, 32] with attention mechanisms [33], could also prove beneficial. These models have demonstrated their effectiveness in language translation and are applicable to time-series data in safety-critical autonomous systems. The evolution of dropout and model ensemble from regularization techniques to tools for uncertainty estimation opens the door for experimentation with other regularization strategies, such as parameter sharing [19], in conjunction with existing uncertainty estimation methodologies.

Further inquiries could also consider the amalgamation of deep ensemble with test-time dropout (i.e., MC dropout) for enhanced uncertainty acquisition. Despite the computational demands, this combination could potentially surpass the performance of Deep Ensemble, MC dropout, and classical methods. Viewing uncertainty estimates research through a regularization lens is promising, aiming to diminish generalization error, thereby bolstering model prediction precision and reliability.

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

## 6 APPENDIX

### 6.1 Derivation

The Derivation of the Negative Log Likelihood is as follows:

$$NLL = -\log \prod_{n=1}^{N} P(x_n|\theta) \tag{2}$$

$P(x_n|\theta)$ is the likelihood

$\theta$ is the parameter of the likelihood function

We substitute the likelihood with the Probability Density Function (PDF) of the Gaussian Distribution because we are dealing with a regression problem.

$$NLL = -\log \prod_{n=1}^{N} \left( \sigma^{-1}(2\pi)^{-\frac{1}{2}} \exp\left( -\frac{(x_n - \mu)^2}{2\sigma^2} \right) \right) \tag{3}$$

Applying the logarithmic addition rule and simplifying it further:

$$NLL = -\sum_{n=1}^{N} \log \left( \sigma^{-1}(2\pi)^{-\frac{1}{2}} \exp\left( -\frac{(x_n - \mu)^2}{2\sigma^2} \right) \right) \tag{4}$$

Since $log(e^x) = x$ and $log(1/a) = -log(a)$, therefore we get as follows:

$$= 0.5 \log(2\pi\sigma^2) + 0.5\sigma^{-2} \sum_{n=1}^{N} (x_n - \mu)^2 \tag{5}$$

[Note, the Deep Ensemble paper [8] doesn't provide the derivation, we have done the derivation for the reader's ease of understanding.]

## 6.2 Dataset and Data Pre-processing

As previously mentioned in the Dataset section, we are using run-to-failure turbo engine data created using the Commercial Modular Aero-Propulsion System Simulation (C-MAPSS), hence there is no missing data. In many real-world scenarios, data may show signs of emerging faults but rarely include complete records of fault progression leading to failure in real-world datasets, hence due to the un-availablility of such real data, we use simulated data.

The dataset comprises several multivariate time series. Each series represents a unique engine of the same model, with "cycle" serving as the time unit and 21 sensor measurements recorded per cycle. It's posited that each engine starts with varying levels of initial wear and manufacturing differences. In this simulated dataset, every engine is presumed to function normally at the beginning of its respective time series. Gradually, as the cycles progress, the engines begin to deteriorate. This degradation intensifies over time, culminating in a critical point when the engine is deemed unfit for further use. Therefore, the final cycle of each series effectively marks the failure point of its engine. For instance, in the provided data sample example Table 3, the engine unit 1 fails at 149 engine cycles whereas engine unit 2 fails at 269 engine cycles respectively.

### 6.2.1 Data Labelling

The data labels have been created as shown in Table 4, the engine unit 1 in Table 3 fails at cycle 149, hence at cycle number 2 the Remaining-Useful-Life (RUL) will be 147 and so on. These labels (i.e RUL) were calculated for all four fleets of the given dataset.

Table 1: Number of Engine Units in each fleet

| Sl. No. | Fleet No. | No. of Engine Units |
|---------|-----------|---------------------|
| 1 | FD001 | 100 |
| 2 | FD002 | 260 |
| 3 | FD003 | 100 |
| 4 | FD004 | 249 |

Table 2: List of Sensor Measurements [22]

| Sensor No. | Description |
|------------|-------------|
| 1 | Total temperature at fan inlet (°R) |
| 2 | Total temperature at LPC outlet (°R) |
| 3 | Total temperature at HPC outlet (°R) |
| 4 | Total temperature at LPT outlet (°R) |
| 5 | Pressure at fan inlet (psia) |
| 6 | Total pressure in bypass-duct (psia) |
| 7 | Total pressure at HPC outlet(psia) |
| 8 | Physical fan speed (rpm) |
| 9 | Physical core speed (rpm) |
| 10 | Engine pressure ratio (P50/P2) |
| 11 | Static pressure at HPC outlet (psia) |
| 12 | Ratio of fuel flow to Ps30 (pps/psi) |
| 13 | Corrected fan speed (rpm) |
| 14 | Corrected core speed (rpm) |
| 15 | Bypass Ratio |
| 16 | Burner fuel-air ratio |
| 17 | Bleed Enthalpy |
| 18 | Demanded fan speed (rpm) |
| 19 | Demanded corrected fan speed (rpm) |
| 20 | HPT coolant bleed (lbm/s) |
| 21 | LPT coolant bleed (lbm/s) |

Table 3: A brief extract from the dataset file for fleet "FD002"

| Engine Unit | Cycle | Operational Setting 1 | Sensor 1 |
|---|---|---|---|
| 1 | 1 | 34.9983 | 449.44 |
| 1 | 2 | 41.9982 | 445.00 |
| 1 | 149 | 42.0017 | 445.00 |
| 2 | 1 | 0.0025 | 518.67 |
| 2 | 2 | 35.0058 | 449.44 |
| 2 | 269 | 42.0047 | 445.00 |
| 260 | 1 | 34.9989 | 449.44 |
| 260 | 2 | 19.9985 | 491.19 |
| 260 | 316 | 35.0036 | 449.44 |

Table 4: An example of dataset labelling for the fleet number 2 data i.e. "FD002"

| Engine Unit | Cycle | Label(RUL) |
|---|---|---|
| 1 | 1 | 148 |
| 1 | 2 | 147 |
| 1 | 3 | 146 |
| 1 | 4 | 145 |

## 6.3 Explanation on Epistemic and Overall-Predictive Uncertainty Computation

The term uncertainty can be described as Epistemic and Overall-Predictive uncertainties. These two uncertainty terms are general concepts and their definition depends on the problem's context. In the same way as Negative Log Likelihood (NLL) is a general term, and its definition is based on the context, for example, in a regression problem the NLL would be substituted with Gaussian Distribution, for a binary classification problem it would be substituted with Bernoulli distribution and for Multi-class classification problem, the categorical distribution (also known as the softmax distribution when used in conjunction with the softmax function in neural networks) would be used [20]. In the same way, the definition of Epistemic uncertainty varies with the problem [9]. Since this paper focuses on a regression model, the two terms are described below according to this context.

In general, for a regression problem, standard deviation/variance is used for uncertainty estimation [9, 34].

The below examples show how to obtain these two uncertainty estimates from the Deep Ensemble model:

Suppose if the ground truth value for one data-point is 20 and if the predictions from the Deep Ensemble of 3 Neural Networks were [18, 19, 21], then the standard deviation of this set [18, 19, 21] is 1.25, which is the epistemic uncertainty. Therefore, this is the uncertainty within the prediction values of the ensemble, this means the individual predictions are in the range of $\pm 1.25$ around the predicted mean.

The Overall-Predictive uncertainty is calculated using the equation in [8]:

$$\sigma_*^2 = [N^{-1} \sum_{n=1}^{N} (\sigma_n^2 + x_n^2)] - \mu_*^2 \qquad (6)$$

$\sigma_*^2$: is the Overall-Predictive variance of the Mixture Gaussian distribution
$\sigma_*$: is the Overall-Predictive standard deviation or Overall-Predictive uncertainty
$N$: is the number of prediction values
$\sigma^2$: is the variance
$\sigma$: is the standard deviation
$x_n$: is the individual prediction value from 1 to N
$\mu_*^2$: is the square of the mean prediction

For the first example mean prediction is 19.33 and square of it is 373.6. Therefore, Overall-Predictive Variance can be calculated as:

$$\sigma_*^2 = [Mean(1 + 18^2, 0.5 + 19^2, 0.5 + 21^2)] - 373.6$$

which gives Overall-Predictive variance as 2.4 and Overall-Predictive standard deviation as 1.54.

Therefore the Overall-Predictive uncertainty is the maximum uncertainty and in this scenario it would be $\pm 1.54$ of the mean prediction. This value is easily interpretable and hence is used as the only metric in the experiment by [12] and [8], however Epistemic uncertainty provides the granular uncertainties [34, 9] within the individual prediction values.

Table 5 below provides two more scenarios, in scenario 3, the uncertainty (Epistemic) is lower within the three predicted values. If the Epistemic uncertainty is not low then the mean prediction will be far from the ground truth.

Table 5: Three scenarios of the uncertainty estimates for the prediction values using Deep Ensemble

| No. | Prediction | Mean | Epistemic $\sigma$ | Overall-Predictive $\sigma$ |
|---|---|---|---|---|
| 1 | [18, 19, 21] | 19.33 | 1.25 | 1.54 |
| 2 | [19.5, 20.5, 20.8] | 20.26 | 0.55 | 0.82 |
| 3 | [15.5, 15.7, 15.9] | 15.7 | 0.16 | 2.156 |

## 6.4 Model Building and Experiment Tables and Figures

Table 6: List of works for uncertainty quantification specifically in Deep Learning models

| No. | Author,Year | Method |
|---|---|---|
| 1 | Sankararaman et. al [35],2013 | Inverse FORM |
| 1 | Charles et. al [36],2015 | Weight uncertainty in neural network |
| 2 | Yarin et al [12], 2016 | Monte Carlo Dropout |
| 3 | Jiri et al [37], 2017 | Variational Gaussian Dropout |
| 4 | Balaji et al [8], 2017 | Deep Ensembles |
| 5 | Zhang et al [38], 2022 | K-means-transformer network |

Table 7: Model Architecture

| Layer Type | Size | Activation | Dropout |
|---|---|---|---|
| Input Size | (Sequence length, Number of Features) (30, 25) | - | - |
| LSTM | Hidden layer: (100 Neurons) | ReLU | 0.2 |
| Dense | Hidden layer: (30 Neurons) | ReLU | 0.2 |
| Dense | output layer - 1 | Linear | - |
| Dense | output layer - 1 | Softplus | - |

Table 8: Parameters/Hyperparameters Settings for Deep Ensemble Model

| No. | Parameter/Hyperparameter | Values |
|---|---|---|
| 1. | Number of epochs | 10 |
| 2. | Batch size | 150 |
| 3. | Sequence Length | 30 |
| 4. | Number of Hidden layers in NN | 2 |
| 5. | Number of nodes in Layer 1 (LSTM) | 100 |
| 6. | Number of nodes in Layer 2 (Dense) | 30 |
| 7. | Optimizer | Adam |
| 8. | Learning rate | 0.01 |

Table 9: Result from Running Deep Ensembles Model on "FD004" (Sequence size as 30 has been used). E represents 'Epistemic' and O-P represents 'Overall-Predictive'.

| No. | Ensemble | Epochs | Nodes L2 | RMSE | $R^2$ | NLL | E std | O-P std |
|---|---|---|---|---|---|---|---|---|
| 1 | 2 | 10 | 30 | 59.27 | -0.24 | 5.11 | 53.99 | 162.05 |
| 2 | 3 | 10 | 30 | 41.91 | 0.38 | 4.30 | 50.17 | 57.97 |
| 3 | 3 | 20 | 30 | 42.90 | 0.35 | 4.43 | 37.02 | 73.61 |
| 4 | 4 | 10 | 30 | 44.16 | 0.31 | 5.62 | 47.8 | 273.7 |
| 5 | 5 | 10 | 30 | 47.21 | 0.21 | 4.52 | 40.85 | 84.11 |

Table 10: Result from Running Deep Ensembles Model (Sequence size 30 has been used). In the below table E represents 'Epistemic' and O-P represents 'Overall-Predictive' uncertainty.

| Fleet No. | No. of Units | Ensemble | Epochs | Nodes L2 | RMSE | $R^2$ | NLL | E | O-P |
|---|---|---|---|---|---|---|---|---|---|
| FD001 | 100 | 3 | 10 | 30 | 30.65 | 0.53 | 3.95 | 35.16 | 42.46 |
| FD002 | 260 | 3 | 10 | 30 | 39.69 | 0.40 | 4.31 | 33.41 | 63.62 |
| FD003 | 100 | 3 | 10 | 30 | 33.21 | 0.50 | 4.28 | 51.77 | 74.54 |
| FD004 | 249 | 3 | 10 | 30 | 41.91 | 0.38 | 4.30 | 50.17 | 57.97 |

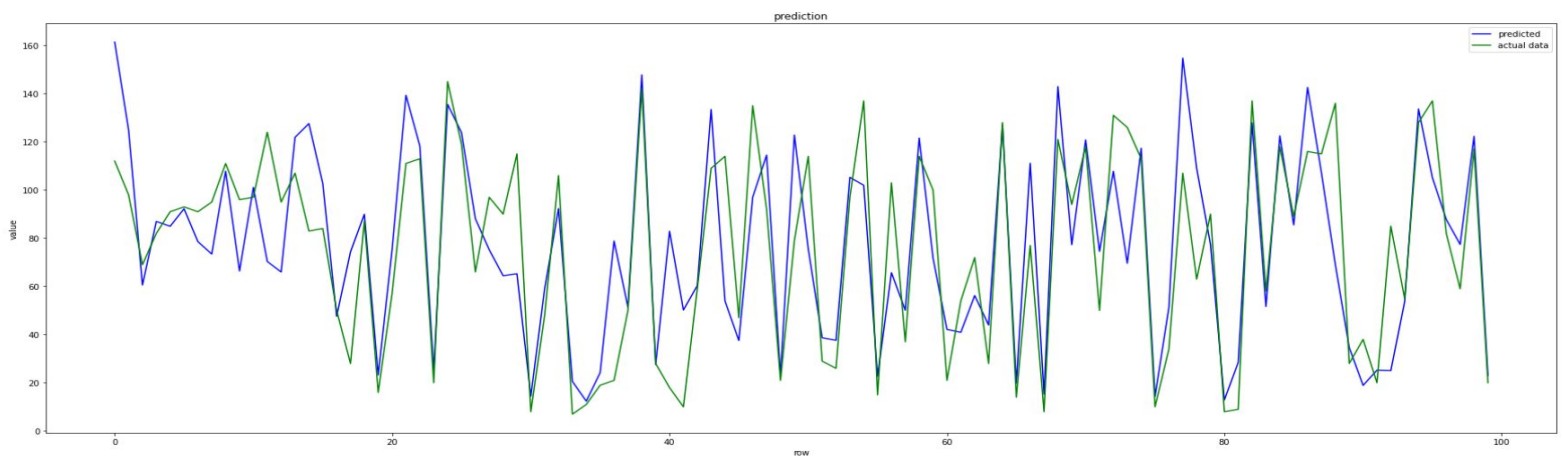

Figure 1: Deep Ensemble LSTM Model for "FD001" fleet dataset showing the actual (green line) and predicted curve (blue line).

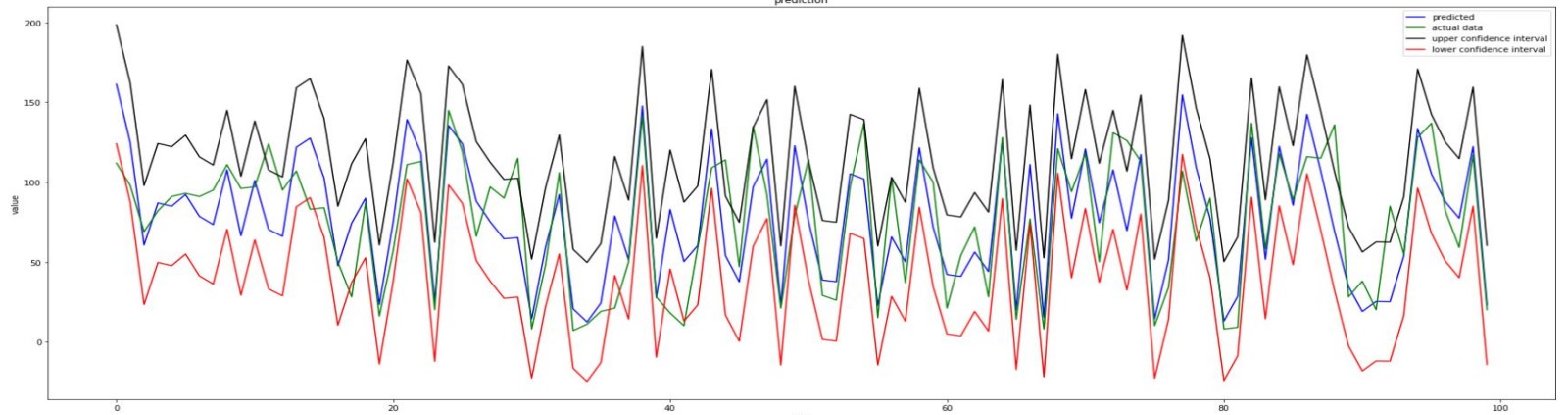

Figure 2: Overall-Predictive Uncertainty for "FD001" data-set using LSTM Ensemble Model showing the actual (green line) and predicted curve (blue line) along with upper (black line) and lower (red line) confidence bound. The confidence interval represents 2 sigma.

