# OpenReview forum: "Remaining-Useful-Life Prediction and Uncertainty Quantification using LSTM Ensembles for Aircraft Engines"
_NeurIPS.cc/2023/Workshop/WANT — WANT@NeurIPS 2023 Poster_

### Official Review · Reviewer_6a41 · 2023-10-23
**Review-UQ-Ensembles-Remaining Useful Life**

**Confidence:** 3

**Review:**

#### **Summary and contributions:**
The paper summarizes the work done on ensembles of LSTM models to predict the remaining useful life (RUL) of aircraft engines.

#### **Strengths:**
The paper has a good literature review on uncertainty quantification methods, with the current state-of-the-art Bayesian approximations (MC dropout, deep ensembles) techniques. The paper aims to estimate uncertainty using deep ensembles for the RUL of engines.

Well written and easy to read. The use of open-source datasets is clearly explained and cited. The authors clearly state software versions, model details, and hyperparameters.

#### **Weaknesses:**

Although LSTM ensembles may not have been previously applied to the RUL prediction problem, other works by Xu, et al (IEEE 2022, vol 71, No. 1), Sankararaman and Goebel (AIAA 2013-1537), Xuan, et al. (Procedia CIRP, 2023, vol 118, 116-121) have been published on uncertainty quantification for the same application, which were not cited in this paper. It would have been essential to compare this paper's approach to the existing literature and what computational benefits would be gained by using LSTM ensembles compared to inverse-FORM (Sankararaman), Gaussian processes (Xu) or the uncertainty-aware framework suggested by Xuan et al.

The authors discuss the importance of decomposing the predictive uncertainty into epistemic and aleatoric uncertainties and report values for both; however, there was no mention of how the uncertainty was decomposed into its respective parts.

#### **Comments/Feedback:**

Figure 1: The caption refers to black and red lines for confidence bounds; however, the lines are not included in the graph. Figure 2 is difficult to read and could be significantly improved by having the confidence interval as a shaded area instead of distinct lines for the upper and lower intervals. Also, there was no mention of what the confidence interval represents (2 sigma or 2.5 sigma?).

Although this work covers a gap relating to the RUL uncertainty quantification with deep ensembles, it does not fall within the scope of this workshop.

#### **Rating:**
3, reject.

---

> ### Author Response · Authors · 2023-11-30
> **Response**
>
> Thank you very much for all the review comments, we appreciate it, these will be addressed in the camera-ready version.
> Just to correct the reviewer's comment - DeepEnsemble is NOT a Bayesian method, yes it is a probabilistic method, but not all probabilistic methods are Bayesian.
>
> The only reason reviewer 6a41 suggested rejection, is because the reviewer considered the paper not a straightforward fit for this Workshop theme. However, our paper is on LSTM-Ensemble, which is built on the work of DeepEnsemble (B. Lakshminarayanan et. al, 2017) and this is an efficient way of getting uncertainty estimates as opposed to heavyweight Bayesian methods, hence it does fit the "Efficiency" criteria of this workshop. The Area Chair does recognise that, and has recommended Accept!!
>
> The reviewer mentioned some literature and suggested comparing our model (LSTM-Ensemble) with methods like Gaussian processes. However, Gaussian Processes are Bayesian methods with very computationally heavy Bayesian inference, our method is non-Bayesian. Moreover, B. Lakshminarayanan et. al (2017) showed that methods like DeepEnsemble outperform Bayesian Methods significantly. Hence, that's why a comparison of LSTM-Ensemble (ours) with the Bayesian method hasn't been included.

---

### Official Review · Reviewer_hXVi · 2023-10-24
**Uncertainty Quantification using Deep Ensembles for Safety-Critical Predictive Models**

**Confidence:** 4

**Review:**

Summary: The paper presents a deep ensemble model for engine health monitoring using Long Short Term Memory (LSTM) network architecture. The dataset used is an open-source dataset provided by NASA, simulating the behavior of turbo engines under different operating conditions. The authors develop a custom ensemble model capable of predicting engine failure and quantifying uncertainty. They evaluate the model's performance using various metrics, including Mean Square Error (MSE), Root Mean Square Error (RMSE), Coefficient of Determination (R2), and Negative Log Likelihood (NLL) for uncertainty quantification.

Strengths of the Paper:

Novel Approach: The paper introduces a novel deep ensemble model for engine health monitoring, addressing a critical problem in predictive maintenance.

Detailed Dataset Description: The authors provide a comprehensive description of the dataset, including the number of engine units, sensor measurements, and the events within each engine cycle. This helps readers understand the context of the research.

Model Architecture: The LSTM-based model architecture is well-documented, and the choice of using LSTM for time series data is explained, which adds to the paper's credibility.

Uncertainty Quantification: The paper focuses on quantifying both Aleatoric and Epistemic uncertainties, providing a valuable insight into the confidence level of the model predictions.

Evaluation Metrics: The authors use a range of evaluation metrics, including RMSE, R2, and NLL, to assess the model's performance comprehensively.

Weaknesses of the Paper:

Lack of Real-World Testing: The study primarily relies on simulation data. It would be beneficial to have real-world data for a more practical evaluation of the proposed model.

Model Complexity: The paper doesn't discuss the potential challenges and complexities associated with deploying the deep ensemble model in real-time industrial applications.

Absence of Baseline Comparisons: The paper doesn't compare the performance of the deep ensemble model with other existing methods or baseline models, making it challenging to gauge its relative effectiveness.

Lack of Interpretability: The paper doesn't delve into the interpretability of the model's predictions, which is essential for decision-making in critical applications.

Limited Discussion of Hyperparameters: While the paper mentions hyperparameters, there is limited discussion about the rationale behind their selection and the impact of tuning.

Code Appendix: The code snippet in the appendix (Appendix 3) is described but not provided, making it difficult for readers to understand the implementation details.

Questions:

What specific types of engine health issues does the deep ensemble model aim to predict and prevent?

How do the uncertainty metrics (Aleatoric and Epistemic) impact the practical use of the model in making maintenance decisions?

What is the significance of the mixture standard deviation, and how does it influence decision-making in engine health monitoring?

What other metrics, apart from RMSE, R2, and NLL, could be relevant for evaluating the model's performance?

Can you explain the choice of LSTM as the underlying architecture and its suitability for time series data?

Is there any consideration of computational efficiency and real-time processing in deploying this model in industrial applications?

How does the model handle sensor failures or missing data in the dataset?

Are there any insights or recommendations for determining the threshold of uncertainty for making maintenance decisions?

What are the practical implications of overconfidence or underconfidence in the model's predictions?

What steps are taken to ensure that the model is aware of its own uncertainty in real-time applications?

---

> ### Author Response · Authors · 2023-11-30
> **Response**
>
> Thank you very much for all the review comments, these will be addressed in the camera-ready version. To respond to the reviewer's comments:
>
> What specific types of engine health issues does the deep ensemble model aim to predict and prevent?
>
> Ans:  - The LSTM-Ensemble model predicts the Remaining-Useful-Life of the Aircraft Engines from the sensor readings (a total of 21 sensors has been used) along with three operational settings and the cycle number.
>
> How do the uncertainty metrics (Aleatoric and Epistemic) impact the practical use of the model in making maintenance decisions?
>
> Ans: - Aleatoric uncertainty, which represents inherent randomness in the data, helps in understanding the noise and variability in sensor readings. Epistemic uncertainty, related to the model's knowledge, can indicate areas where the model might need more data or training. Together, they inform maintenance decisions by highlighting the confidence level of the model's predictions and identifying areas needing further data collection or model refinement.
>
> What is the significance of the mixture standard deviation, and how does it influence decision-making in engine health monitoring?
>
> Ans: - Here in our paper, Mixture standard deviation is reffering to the overall 'predictive uncertainty'. To avoid confusion, the term Mixture is updated with 'overall predictive' uncertainty. In the engine health monitoring model, a higher standard deviation suggests lower confidence, influencing maintenance decisions by potentially prompting earlier or more thorough inspections.
>
> What other metrics, apart from RMSE, R2, and NLL, could be relevant for evaluating the model's performance?
>
> Ans: - The paper uses the RMSE, R2, and NLL measures for evaluation. For even further evaluation, metrics like Mean Squared Prediction Error (MSPE) or Mean Absolute Scaled Error (MASE) might be useful, which can be explored as further work.
>
> Can you explain the choice of LSTM as the underlying architecture and its suitability for time series data?
>
> Ans: - The reason LSTM networks are used is due to their suitability for time series data as they can capture long-term dependencies and patterns over time. This makes them ideal for predicting RUL, where historical sensor data and past
> performance trends are crucial for accurate forecasting.
>
> Is there any consideration of computational efficiency and real-time processing in deploying this model in industrial applications?
>
> Ans: - In industrial applications, models need to balance accuracy with computational efficiency. Techniques like model simplification, quantization, or using efficient architectures are common. Keeping the computational efficiency aspect in mind, we proposed the LSTM Ensemble method, which is way more computationally efficient as compared to Bayesian methods, what has been previously used in uncertainty quantification.
>
> How does the model handle sensor failures or missing data in the dataset?
>
> Ans: - As previously mentioned on the Dataset section, we are using run-to-failure turbo engine data created using the Commercial Modular Aero-Propulsion System Simulation (C-MAPSS), hence there is no missing data. Since, in many real-world scenarios, data may show signs of emerging faults but rarely include complete records of fault progression leading to failure in real-world datasets, hence due to the availablility of such real data, hence simulated data has been used.
>
> Are there any insights or recommendations for determining the threshold of uncertainty for making maintenance decisions?
>
> Ans: -  Determining the threshold for uncertainty in maintenance decisions is context-dependent and involves risk assessment by the organisations. This paper does not detail specific thresholds but highlights that lower NLL values indicate more reliable uncertainty estimates. Maintenance teams might set thresholds based on acceptable risk levels and the criticality of engine components.
>
> What are the practical implications of overconfidence or underconfidence in the model's predictions? What steps are taken to ensure that the model is aware of its own uncertainty in real-time applications?
>
> Ans:-
> Overconfidence can lead to missed maintenance and potential failures, while underconfidence might result in unnecessary maintenance and increased costs. Both impact the reliability and cost-effectiveness of maintenance strategies.
> In regards to ensuring the model is aware of its own uncertainty in real-time applications, this can be achieved through continuous monitoring of the LSTM-Ensemble model and incorporating feedback mechanisms.
>
> Code: Its available in the GitHub page - https://github.com/odeb1

---

### Official Review · Reviewer_TWZz · 2023-10-26
**Wwhile the paper tackles an important problem and offers a methodological approach, it requires further technical depth and rigor to meet the standards of high-quality academic publication.**

**Confidence:** 4

**Review:**

Abstractness of Model Specification
The paper employs Long Short-Term Memory (LSTM) networks in a deep ensemble model for time-series prediction. However, the manuscript does not offer detailed justification as to why LSTMs were chosen over other time-series models such as ARIMA, GARCH, or even simpler RNNs. Given that the paper aims for a deep ensemble model, a comparative analysis with other architectures would enrich the manuscript and provide stronger justification for the chosen model.

Uncertainty Quantification
The paper makes an attempt at uncertainty quantification via standard deviation outputs, but it lacks comprehensive empirical or theoretical justification for this approach. The standard deviation is indirectly supervised by the loss function NLL (Negative Log Likelihood). However, the paper does not provide evidence on how effective this indirect supervision is in reflecting the model’s uncertainty.

Dropout Regularization
The manuscript delineates the use of dropout after each LSTM layer to prevent overfitting. However, it fails to elaborate on the selection process for the dropout rate, which is a crucial hyperparameter. The manuscript could be strengthened by providing sensitivity analyses for dropout rates and their impact on overfitting.

Hyperparameter Optimization
The paper briefly mentions that multiple experiments with different values of hyperparameters were conducted. However, the manuscript does not provide details on how the hyperparameters were tuned or optimized. Were grid search, random search, or Bayesian optimization techniques used? This is a significant omission given the impact of hyperparameter tuning on model performance.

Software and Data
The paper uses Python’s Keras library and TensorFlow as a backend, along with some open-source codes. While using open-source codes and libraries is not a problem per se, justifying why these specific tools were chosen over others would add value. Moreover, the manuscript does not provide any insights into the data preprocessing steps undertaken before feeding the data into the LSTM model.

Scalability and Computational Cost
The paper does not discuss the computational cost of training the deep ensemble model, an important consideration especially for larger datasets or real-time applications. Understanding the computational efficiency or limitations of the model is crucial for assessing its practical applicability.

Lack of Benchmark Comparisons
The paper does not provide any comparative analyses with existing models or techniques, either in terms of accuracy or computational efficiency. Such comparisons are vital for establishing the relevance and superiority of the proposed model.

Novelty and Contribution
The paper utilizes existing LSTM and deep ensemble techniques but does not introduce any significant novel contributions to the field. Given the state-of-the-art in deep learning and ensemble models, the paper would greatly benefit from a section discussing the novelty or the specific challenges that this model aims to address.

Relevance and Justification
Given the ubiquity of time-series data and the growing interest in uncertainty quantification, the paper’s topic is certainly relevant. However, the manuscript could substantially increase its impact by addressing the aforementioned technical gaps. A rigorous mathematical justification for the model's architecture and the uncertainty quantification methods used would further solidify its relevance and contribution to the field.

---

> ### Author Response · Authors · 2023-12-01
> **Response**
>
> Thank you very much for all the review comments, these will be addressed in the camera-ready version. Moreover, thank you for acknowledging that our paper's topic is certainly relevant for this workshop.
>
> Some of the points made by this reviewer "TWZz" certainly contradict the points made by the below reviewer "hXVi", who appreciates these points such as ("Novel Approach: The paper introduces a novel deep ensemble model for engine health monitoring, addressing a critical problem in predictive maintenance.
> Model Architecture: The LSTM-based model architecture is well-documented, and the choice of using LSTM for time series data is explained, which adds to the paper's credibility.
> Uncertainty Quantification: The paper focuses on quantifying both Aleatoric and Epistemic uncertainties, providing valuable insight into the confidence level of the model predictions.
> Evaluation Metrics: The authors use a range of evaluation metrics, including RMSE, R2, and NLL, to assess the model's performance comprehensively.")
>
> There is inconsistency in the reviewer's comment, anyway, the response to the reviewer (TWZz) is as follows:
>
> Justification for the methods used for Remaining-Useful-Life (RUL) and uncertainty quantification:
> LSTM with Deep Ensemble has been used over other methods such as ARIMA, GARCH, or even simpler RNNs, is because ARIMA and GARCH are primarily linear models and might not capture non-linear patterns effectively, given the fact that our dataset is non-linear. While simple RNNs can handle non-linearities, they are not as efficient as LSTMs in learning from complex, high-dimensional data. LSTMs, being a more advanced form of RNNs, can model complex non-linear relationships in data much better.
>
> Moreover, LSTMs are more robust to noise and non-stationarity in data compared to ARIMA and GARCH. Although ARIMA can handle some level of non-stationarity, LSTMs, especially when used in an ensemble setting, can better adapt to changing patterns in the data without the need for stringent pre-processing like stationarity tests and transformations.
>
> While GARCH models are specifically designed to model volatility clustering and heteroskedasticity in financial time series, LSTMs can also capture these features, especially when trained on large datasets. The ensemble approach further enhances this capability by aggregating insights from multiple models. The Deep Ensemble approach, when combined with LSTMs, can lead to improved prediction accuracy. Each model in the ensemble will capture different aspects of the data, and their aggregation leads to a more accurate and robust model. This is particularly useful in our scenarios because the data contains complex patterns that a single model will not be able to capture effectively.
>
> By integrating LSTMs, which effectively capture temporal dynamics, with Deep Ensembles, the model not only predicts future values but also gives a more reliable estimate of the uncertainty associated with these predictions. This is crucial for decision-making processes in fields where understanding the confidence in predictions is as important as the predictions themselves.
>
> Finally, LSTMs, especially when implemented with efficient computational frameworks, can be scaled for large datasets and real-time analysis, which is a challenge for traditional models like ARIMA and GARCH which require extensive computations for each new prediction.
>
> Justification for Uncertainty Quantification Approach:
> To quantify uncertainty, models in an Ensemble are designed to output probabilistic predictions, so in this regression task, the model predicts a mean and variance for each output. The ensemble aggregates these probabilistic outputs. The variance (or standard deviation) across the ensemble’s predictions provides an empirical estimate of uncertainty. The NLL loss function is particularly suited for models that output probabilistic predictions, the outputs represent parameters of a probability distribution (e.g., the mean and variance of a Gaussian distribution).
> Regarding the indirect supervision of uncertainty - when the model is trained using NLL, it adjusts not only the prediction (e.g., mean) but also the associated uncertainty (e.g., variance). Hence, minimizing NLL inherently involves correctly estimating the variance – if the variance is over or underestimated, the NLL will be higher. B. Lakshminarayanan et. al (2017) showed empirical evidence that when models are trained with NLL show better-calibrated uncertainty estimates.
>
> Comment on Dropout Regularization and Hyperparameter Optimization:
> We implemented a grid search where several models are trained with different dropout rates (such as 0.1, 0.2, 0.3, ..., 0.7) and compared their performance on a validation set. 0.2 is found to be the most optimal value for the dropout rate.

---

> ### Author Response · Authors · 2023-12-01
> **Response**
>
> Software and Data:
> Further info on data preprocessing has been added as follows: The data labels have been created as shown in Table 5, the engine unit 1 in Table 4 fails at cycle 149, hence at cycle number 2, the Remaining-Useful-Life (RUL) will be 147 and so on. These labels (i.e RUL) were calculated for all four fleets of the given dataset.
>
> In terms of using open-source Deep Ensemble code, since the Deep Ensemble paper by B. Lakshminarayanan et. al (2017) is already been published and has some open-source code available, hence rather than re-inventing the wheel, we have used a snippet of the open-source code, this has already been cited properly in the paper. However, the implementation of the LSTM-Ensemble model is done by us.
>
> Novelty and Contribution:
> We proposed the "LSTM-Ensemble" technique in building a regression model to predict the Remaining-Useful-Life (RUL) of aircraft engines along with uncertainty quantification of the model, utilizing the well-known run-to-failure turbo engine degradation dataset. We addressed the overlooked yet crucial aspect of uncertainty estimation in previous research, and we revamp the LSTM architecture to facilitate uncertainty estimates, employing Negative Log Likelihood (NLL) as the training criterion.
>
> Benchmark Comparisons:
> The reviewer suggested comparing our model (LSTM-Ensemble) with other methods. But, to the best of our knowledge, this is a pioneering work in this application domain from a non-Bayesian approach. Other methods like Gaussian processes have been explored for this application domain, but Gaussian Processes are Bayesian methods with very computationally heavy Bayesian inference, our method is non-Bayesian. And B. Lakshminarayanan et. al (2017) have already shown that methods like DeepEnsemble outperform Bayesian Methods significantly. That's why a comparison of LSTM-Ensemble (ours) with the Bayesian method hasn't been included.

---

### Meta-Review · Area_Chair_JChe · 2023-10-27

**Recommendation:** Accept (Poster)
**Confidence:** 4

**Metareview:**

There is a lot of valid criticism for this paper. Also, the fit to the topic of the workshop is not straightforward. However, I find it important to present such a practice-inspired works at the workshop, and also expect that the authors can benefit a lot from the participation. I recommend the authors to thoroughly consider the remarks by reviewers while preparing the poster and for future submission of the work.

---

> ### Author Response · Authors · 2023-12-02
> **Response**
>
> First of all, thank you very much for the review. Just to mention, there have been inconsistencies in the reviewer's comments. Just to correct the reviewer's statement - DeepEnsemble is NOT a Bayesian method, yes it is a probabilistic method, but not all probabilistic methods are Bayesian.
> The camera-ready version has been updated to address all the review comments. Thank you once again for your time.

---

### Decision · Program_Chairs · 2023-10-28

**Decision:**

Accept (Poster)

**Comment:**

We thank the authors for their time and contribution to WANT and we are pleased to share that after the reviewing process the paper has been accepted. Congratulations! We encourage the authors to consider reviewers' feedback for the improvement of the camera-ready version. We hope to see you in person at the workshop and brainstorm on efficient training research together!

---

> ### Author Response · Authors · 2023-12-02
> **Response**
>
> Thank you Program chairs for your "Accept" decision. We appreciate your time and thank you for the review.